# Effective Dose Estimation in Computed Tomography by Machine Learning

**DOI:** 10.3390/tomography11010002

**Published:** 2025-01-02

**Authors:** Matteo Ferrante, Paolo De Marco, Osvaldo Rampado, Laura Gianusso, Daniela Origgi

**Affiliations:** 1Medical Physics Unit, IEO, European Institute of Oncology IRCCS, 20141 Milan, Italy; matteo-ferrante@hotmail.it (M.F.); paolo.demarco@ieo.it (P.D.M.); 2Medical Physics Unit, A.O.U. Città della Salute e della Scienza di Torino, 10126 Turin, Italy; orampado@cittadellasalute.to.it (O.R.); lgianusso@cittadellasalute.to.it (L.G.)

**Keywords:** artificial intelligence (AI), patient radiation protection, dose tracking

## Abstract

Background: Computed tomography scans are widely used in everyday medical practice due to speed, image reliability, and detectability of a wide range of pathologies. Each scan exposes the patient to a radiation dose, and performing a fast estimation of the effective dose (E) is an important step for radiological safety. The aim of this work is to estimate E from patient and CT acquisition parameters in the absence of a dose-tracking software exploiting machine learning. Methods: In total, 69,037 CT acquisitions were collected with the dose-tracking software (DTS) available at our institution. E calculated by DTS was chosen as the target value for prediction. Different machine learning algorithms were selected, optimizing parameters to achieve the best performance for each algorithm. Effective dose was also estimated using DLP and k-factors, and with multiple linear regression. Mean absolute error (MAE, mean absolute percentage error (MAPE), and R^2^ were used to evaluate predictions in the test set and in an external dataset of 3800 acquisitions. Results: The random forest regressor (MAE: 0.416 mSv; MAPE: 7%; and R^2^: 0.98) showed best performances over the neural network and the support vector machine. However, all three machine learning algorithms outperformed effective dose estimation using k-factors (MAE: 2.06; MAPE: 26%) or multiple linear regression (MAE: 0.98; MAPE: 44.4%). The random forest regressor on the external dataset showed an MAE of 0.215 mSv and an MAPE of 7.1%. Conclusions: Our work demonstrated that machine learning models trained with data calculated by a dose-tracking software can provide good estimates of the effective dose just from patient and scanner parameters, without the need for a Monte Carlo approach.

## 1. Introduction

Computed tomography is one of the most widely used diagnostic imaging techniques, due to its speed and ability to detect anatomical details that are fundamental to the diagnosis of numerous pathologies. However, it is based on the use of ionizing radiation, a possible source of risk for the patient. The estimation of the effective dose (E) administered during radiological investigations of computed tomography in the medical field is of particular interest for radiation protection. According to ICRP 147 [1], the effective dose is used in medicine for comparing doses from different medical procedures, informing judgements on justification, and establishing constraints for carers and volunteers in medical research.

There are several approaches to the estimation of this parameter, ranging from the simplest, in which it assumes a linear relationship between Dose Length Product (DLP) and effective dose, to more complicated approaches based on Monte Carlo simulations on mathematical phantoms. In particular, there is dose-tracking software that automatically queries the hospital’s PACS and retrieves information from the structured dose report and the image itself. A great deal of this software includes Monte Carlo simulations on a set of mathematical anthropomorphic phantoms. They include, for example, representative phantoms of men and women of different sizes, ages, and situations, such as ones dedicated to pregnant women. The scout image of the patient is selected and the most similar phantom is associated with the patient, and then the values of the simulation are corrected through the use of look-up tables, which provide the corrective effective dose values to the different organs according to the acquisition parameters.

The range of possible approaches is very wide and different software implements multiple solutions, with different levels of complexity, ranging from estimates based on coefficients to the use of the same images acquired.

The primary aim of our work is to explore the power of machine learning in effective dose estimation, starting from values obtained using a commercial dose-tracking software (Radimetrics, Bayer Healthcare [2]). Such software provides effective doses using some patient information like age, gender, height, and weight, the acquisition parameters, and the dose indices provided by the acquisition devices.

The secondary aim of our work is to predict the right dose class with high accuracy, starting from effective dose estimates. The new Italian regulation (D.Lgs. 101/20), a transposition of the Euratom 59/2013, establishes that every radiological procedure must indicate in the radiological report information relating to patient exposure in terms of classes of effective dose (E). There are four possible dose classes: class I if E < 1 mSv, class II if 1 < E < 5 mSv, class III if 5 < E <1 0 mSv, and class IV if E > 10 mSv.

## 2. Materials and Methods

We collected and anonymized dose-tracking information and developed a pipeline in Python to preprocess them and feed the data to machine learning models, to estimate the effective dose.

The pipeline explained hereafter is summarized in Figure 1.

Most of this work was carried out using the scikit-learn (https://scikit-learn.org/stable/ accessed on 20 March 2021) [3] and tensorflow (https://www.tensorflow.org/ accessed on the 20 March 2021) [4] Python libraries.

### 2.1. Data

#### 2.1.1. Internal Dataset and Variable Description

We collected 69,037 CT acquisitions of two different devices of the same vendor from January 2019 to February 2021 and sent them to the commercial dose-tracking software (Radimetrics, Bayer Healthcare). This software is based on mathematical phantoms coupled with precomputed Monte Carlo and produces an estimate of the effective dose for each acquisition.

The output of this software was used to explore the performances of several machine learning models and whether they are able to predict the effective dose just by using patient and scanner information, to produce a model able to estimate the dose and the class dose just by using the information available in the header of the DICOM image.

We collected a great deal of information from the dose-tracking software, including descriptions, parameters, and more. We chose to feed our model with a subset of the parameters that were exportable from Radimetrics^TM^. In particular, we included in our model three different types of information:

Patient-related information: gender, weight, height, and age.

Scanner-related parameters: kVp, minimum, maximum, and mean value of mAs, nominal mA, rotation time, pitch and collimation.

Dose estimate indices: CTDI_vol and DLP for both head and body phantoms.

Patient-related information is useful to include corrections about patient size.

Scanner-related parameters are useful to include corrections about modulation to follow patient shape and attenuation.

Dose estimate indices provide a baseline value because of the well-known linear relationship between DLP and E. These indices are computed based on the values computed on two types of uniform water phantom, head and body, with, respectively, 16 and 32 cm of diameter.

#### 2.1.2. External Dataset

To ensure the generalization capability of our best model, we also employed a dataset from a different institution with a different dose-tracking software (Physico, Emme Esse, Italy) [5], collecting 3800 acquisitions from another hospital (Azienda Ospedaliero Universitaria (A.O.U.) Città della Salute e della Scienza di Torino).

### 2.2. Preprocessing

We preprocessed the data exported from Radimetrics^TM^ developing a pipeline able to reduce both the features and the outliers present in this dataset.

We removed all samples that included missing values and we excluded features like series description and hospital-related information, to anonymize data and reduce hospital custom name dependence. This process simplified the dataset for an eventual generalization procedure on external datasets. At the end of the preprocessing procedure, the result was a cleaned dataset of over 69,000 samples, that was split into training (70%) and test (30%).

The dataset included 16 features describing the acquisition procedure and the target variable, i.e., the effective dose, estimated by Radimetrics^TM^ using the mathematical phantoms and look-up tables with the definition of effective dose as in the ICRP 103 [6].

### 2.3. Exploratory Data Analysis

To be sure that machine learning models could learn the right correlation between data and outputs, we explored our data to gain insights of the most important features.

As a first step, we computed a clustermap of the features in the training set using the Python seaborn [7] library. We also computed a correlation map (see Figure 2) between features and the target variable, effective dose, defined as “ICRP_103_mSv” in Radimetrics^TM^.

We can observe that is possible to cluster together scanner features and dosimetric indices because they are highly correlated, containing information about the parameters of the scanning procedure and the expected dosimetric output. In Figure 3, we can observe the correlation between the features in the training set and the target variable. We can observe that DLP (Body) shows a high correlation value, as expected, so we can assume that a good model will start with a nearly linear estimate using DLP and adjust this baseline value including higher order dependencies from other variables.

To complete our exploratory analysis, we also included a Principal Component Analysis (PCA) to check which features affect mostly the variability in the output. We used the scikit-learn implementation of PCA with default parameters to obtain a ranking of the features based on their importance in the variability explanation. In Figure 4, it is possible to observe this parameter in the PCA decomposition. The most important feature is DLP (Body) that can explain 73% of the total variance in the dataset. The second most important variable is DLP (Head) that accounts for about 23% of the variance in the dataset.

### 2.4. Model Comparison

In our study, we compared three machine learning algorithms: a neural network [8], a support vector machine [9], and a random forest [10].

For each of these models, the request was to perform a regression and predict an estimate of the effective dose starting from the features in the training dataset.

Since machine learning model performance is generally sensitive to hyperparameter values, we used the Weight & Biases (W&B) [11] library to experiment with different combinations and choose the optimal hyperparameters for this particular task.

For all optimization studies, we used the training dataset, further divided into a part for training and a part for validation with a 70–30% division.

For all the models, we explored different combinations of hyperparameters with a random search, collecting the results in a series of reports in order to study the impact of each parameter on the final results and compare the different models with each other.

#### 2.4.1. Neural Network

A neural network is a mathematical model where a network of neurons (computational units) connected together is trained by adjusting the values of the weights in order to minimize the value of a cost function that represents a measure of the errors between the network estimates and the real values. In a regression network, the last computational unit proposes a regression that estimates the final value.

We developed a simple script that generates the architecture of a neural network using the high-level API keras [8] to build a multilayer perceptron.

In our optimization analysis, we included these possible combinations of hyperparameters, that are summarized in Table 1:

The “hidden_layers” parameter is a list where the number of elements means the number of hidden layers and the value of the ith item of the list is the number of computation units in the ith layer (for example, [256, 128, 64] means 3 hidden layers with, respectively, 256, 128, and 64 units).

The “do” parameter is the Drop-Out factor, namely, the percentage of connections that are cut out at each layer.

The “activation” parameter is the activation function used to introduce non-linearity in the network. We used Rectified Linear Unit, “relu”.

The “solver” parameter is related to the optimizer choice.

The “epochs” parameter is the number of times that the whole training dataset is shown to the network.

The “batch_size” parameter is the number of samples that are shown at the same time to the network.

The “init_lr” parameter controls the starting learning rate of the optimizer algorithm.

For all parameters, up to four choices are provided. The W&B library randomly picks 30 different configurations to study the impact of these hyperparameters on final performances.

#### 2.4.2. Random Forest Regressor

Random forest models average the prediction of a high number of decision trees. Each decision tree tries to split the dataset iteratively and starting from the mean value in the dataset, adjust the regression to output an estimate of the effective dose based on the values of the different features. Each tree is trained using just some of the total features and part of the dataset in a parallel way in order to reduce overfitting chance. We used the random forest regressor implementation in the scikit-learn library [10].

In our optimization analysis, we included these possible combinations of hyperparameters, that are summarized into Table 2:

‘n_estimators’ is the number of decision trees to train.

‘max_depth’ is the maximum number of splits allowed for each tree.

‘ccp_alpha’ is a regularization parameter that is involved in the pruning procedure following the Minimal Cost-Complexity Pruning procedure.

For all parameters, up to four possible choices are provided and the W&B library randomly picks up to 15 different configurations to study the impact of these hyperparameters on final performances.

### 2.5. Model Selection and Training

For each configuration of each model, we evaluated three parameters, namely, the r2 score, the mean absolute error (MAE), and the mean absolute percentage error (MAPE), considering the estimates from Radimetrics as our gold standard. All models were evaluated using these metrics and the best configuration of each algorithm was chosen to fit the data.

We fitted the best model with the whole training set and evaluated the performances on the test set and the external dataset, with an increasing number of features, as shown in Table 3.

Then, we finally chose the best-performing model and used it to perform a class dose prediction.

### 2.6. Effective Dose from Other Estimates

To evaluate the added value of effective dose estimates by machine learning, effective doses were also evaluated using DLP and k- factors from the work of Romanyukha et al. [12]. Scan regions were attributed according to study description, and correspondent k-factors were used for E estimate, according to the well-known formula
E = k × DLP

In addition, E was evaluated with multiple linear regression using the 6 features of Table 4 (gender, weight, height, CTDI, DLP, and scan region), computing the R^2^ of the estimates.

MAE and MAPE were calculated accordingly, considering the estimates from the dose-tracking software as our gold standard.

## 3. Results

### 3.1. Internal Validation

In the optimization phase, we logged a lot of information about the impact of the selected hyperparameters on the performance of the model. Weights & Biases provide a lot of useful panels to explore correlations and results of different runs (see Figure 5 as an example).

For the neural network, in the optimization phase, MAE ranged from 0.57 mSv to 2.99 mSv, with an R2 score that ranged from 0.36 to 0.97. Analyzing the W&B report, we chose as final hyperparameters the combination of hidden_layers = [512, 256, 128, 64], do = 0, init_lr = 1 × 10^−4^, solver = ‘adam’, and batch_size = 512.

For the support vector machine, in the optimization phase, MAE ranged from 0.68 mSv to 4 mSv, with an r2 score that ranged from 0.14 to 0.96. Analyzing the W&B report, we chose as final hyperparameters the combination of C = 1, kernel = “rbf”, and gamma = “scale”.

For the random forest regressor, in the optimization phase, MAE ranged from 0.44 mSv to 0.7 mSv, with an r2 score that ranged from 0.96 to 0.98.

We fitted all the selected models with the whole training dataset and then evaluated their performances using mean absolute error, mean absolute percentage error, and the coefficient of determination. Table 4 summarizes the results of each model. We found good r2 with all models and the overall best-performing model was the random forest regressor, which achieved the lowest mean absolute error.

The visual correlation between predicted and expected doses for random forest is shown in Figure 6.

### 3.2. Dose Class Prediction

The new Italian regulation (D.Lgs. 101/20), a transposition of the Euratom 59/2013, establishes that every radiological procedure must indicate in the radiological report information relating to patient exposure in terms of classes of effective dose (E).

We chose our best model, the random forest regressor, and mapped all the Radimetrics and model estimates of the effective dose into the corresponding dose class to compute accuracy, precision, recall, and f1-score for each class to ensure that our model is able to reasonably place each acquisition in the right dose class. We achieved high results for each class, which further supports the hypothesis that random forest could be a way to estimate the effective dose and predict the right dose class (Table 5).

### 3.3. External Validation

We reached good results on test data coming from the same source, so devices, body regions, and kind of exam are really homogeneous between training and test data (Table 6 and Table 7). To further check the generalization ability of our model, we also collected data coming from different hospitals, with similar body regions (mainly body and head exams).

### 3.4. Interpretability

Assuming that the Monte Carlo approach is the gold standard for effective dose estimation and that dose-tracking software is the tool that can provide us with the best reasonably achievable estimate of this parameter, the use of simple machine learning models can also be a useful tool when trying to explore how these models use information to estimate the variable of interest, without having access to the image. Surely, it is necessary and inevitable that some biases present in the data are included and learned by the model—and this is probably the cause of the different performances if evaluated on data from other sources—and for models such as random forest, it is possible to study the weight of each feature and also the trend of the most important ones.

For this purpose, we calculated the importance of the features (see Figure 7), where we found results in agreement with what was expected from the preliminary analysis based on the correlations and the decomposition into main PCA components; specifically, we identified DLP Body as the most important feature from which an initial estimate starts, which is then adjusted using the other features to account for specific variables.

To further explore the relationships learned from the model, we developed a partial dependency analysis for three variables of particular interest: DLP, mean mAs, and kVp.

The results are shown in Figure 8: these plots are a measure of the trend of the prediction as the only variable indicated varies, perturbing the input data, and plotting how the estimate varies according to the parameter studied.

We have identified an almost linear trend for DLP and a low dependence (a practically horizontal line at the height of the average effective dose value in our dataset—6.88 mSv), in agreement with the fact that DLP (Body) is the parameter that most affects the estimate.

## 4. Discussion

Estimating the effective dose is a crucial point for radiation protection and the optimization of diagnostic processes based on medical imaging with ionizing radiation. Models capable of accurately estimating the effective dose are the first step to understand which parameters have the greatest impact on the dose delivered, linked to possible risks for the patient. ICRP 147 underlines the importance of the effective dose in medical radiation exposure and the need to standardize effective dose calculations to provide reference dose coefficients for specified radiographic and CT procedures.

The definition of the effective dose includes the study of the dose to the organs, weighted by the relative coefficients that take into account the different radiosensitivity of the tissues. Different calculation methods, different software, and different investigation methodologies have an important impact on the calculation of the dose. As demonstrated by De Mattia et al. [13], there are important differences in different software, with variability in the estimate of the effective dose up to 35% and in other parameters beyond 100%. According to the authors of this work, the coefficients found in the literature and implemented in the software are not always representative of the adequate values for the different clinical protocols in different facilities.

Ultimately, despite the clear certainty of correlations with some variables such as dosimetric indices, kVp, and current modulation, differences in scanners and in patients’ characteristics can lead to very important differences in effective dose estimates.

Aware of these limitations, we explored the performance of different machine learning algorithms, trained on a very large and homogeneous dataset, to obtain a fast and light model capable of estimating the effective dose with reasonable accuracy starting from the patient’s anatomical data and information tied to the scanner, without the need to process the image.

The results of the optimization phase were quite expected: for example, the architecture of the neural network and the choice of the best hyperparameters are reasonable, given the type of data to be handled. For the support vector machine, ‘rbf’ kernel maps input data into higher-dimension space, allowing better generalization, while for random forest, a small value of ‘max depth’ (10) and an intermediate number of ‘n_estimators’ (500) are able to capture the complexity of the dataset, allowing a good generalization without overfitting.

On test data coming from the same hospital, the random forest regressor showed really good predictions, with a mean absolute error below 0.5 mSv and an average accuracy of 0.96 in the prediction of the dose class.

The same model showed the ability to incorporate all the essential correlations to produce a reasonable estimate and to reproduce the results with a low average percentage error, even on cases from different sources, with patients, scanners, and clinical protocols that certainly differ, as shown from the results of the external validation, with an absolute error well below 1 mSv and a percentage error below 10%.

The prediction of dose class for the external dataset resulted in an accuracy of 0.95.

The performance of the model proved to be sufficiently robust to account for the fact that two dose-tracking software estimated organ doses, and thus effective doses, using different phantoms.

Such results could lead to a development of a pipeline in which features from Table 4 could be extracted from each acquisition, to feed the model, obtaining the effective dose estimates. The features needed are, in fact, reported in the DICOM metadata of each CT acquisitions, thus being readily available and easily implemented in an automatic workflow.

### Limitations

While the random forest regressor model performed very well on data from our center and showed promising results even on data from different centers, this type of approach includes several limitations. As already described, different scanners and protocols lead to great variability in the final estimates even on the same software, which are amplified if different software is used for the estimation. Machine learning is promising for this type of study because, as we have shown, it is able to accurately learn correlations and even expected physical trends; however, training on data from a limited number of scanners and from a single center induces the model to learn the specific biases of that center and of those scanners, which worsens the performance of generalizations on different situations. Undoubtedly, extensive training with data from multicentric studies could result in an estimate that is less affected by this type of problem and with a greater capacity for generalization.

## 5. Conclusions

In this work, we presented an exploration of how machine learning could help predict the effective dose in CT imaging. We proposed and evaluated different optimized models and found the best performances with random forest regressor, which could be a tool to estimate the effective dose and thus the effective dose class with high accuracy once trained on enough data.

We also checked that the model had correctly learned the physical trends that are expected from the literature and checked it against datasets coming from different centers, using different devices, protocols, and even dose-tracking software.

We found results with a mean absolute error and a mean absolute percentage error in line with those expected in the literature for that type of difference.

We can conclude that machine learning could have an increasing role in the future of effective dose estimation and could perhaps be included in dose-tracking software after extensive training on multicentric data. This could soon be a new tool for personalized, lightweight effective dose estimation to be performed online just after acquisition.

## Figures and Tables

**Figure 1 tomography-11-00002-f001:**
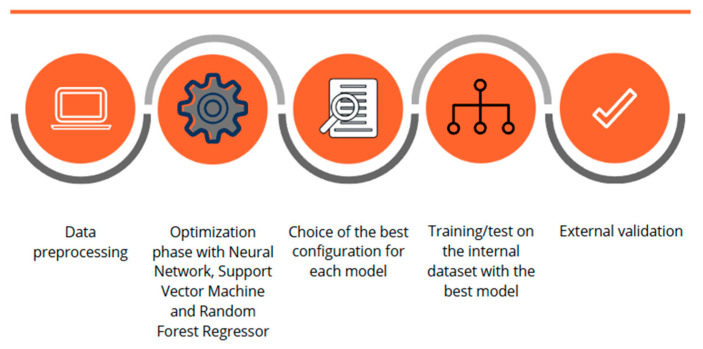
Pipeline for effective dose estimation.

**Figure 2 tomography-11-00002-f002:**
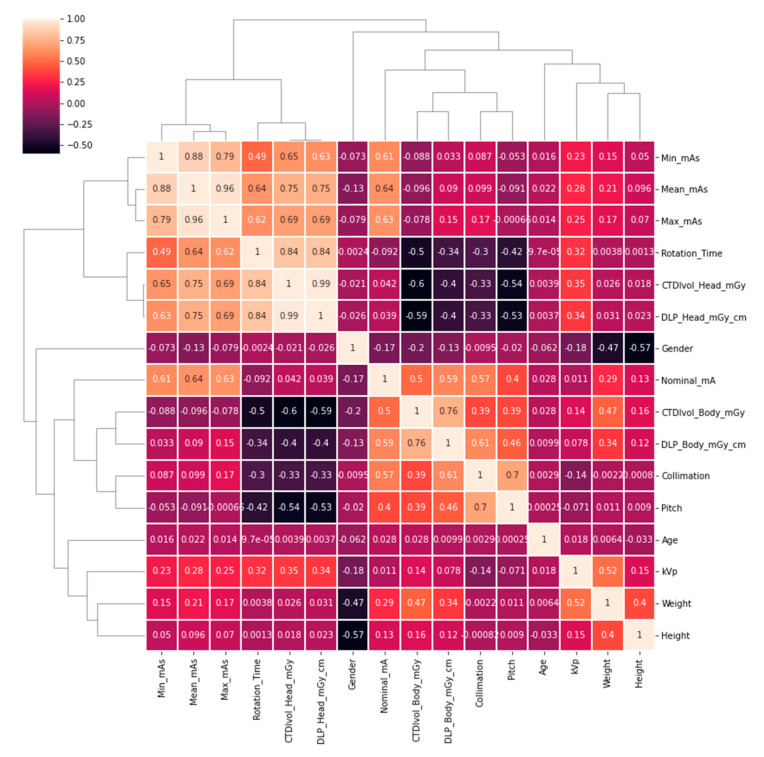
Cluster map computed from the correlation between features in the training dataset. We can observe that features can be clustered into patient information, scanner parameters, and dose descriptors.

**Figure 3 tomography-11-00002-f003:**
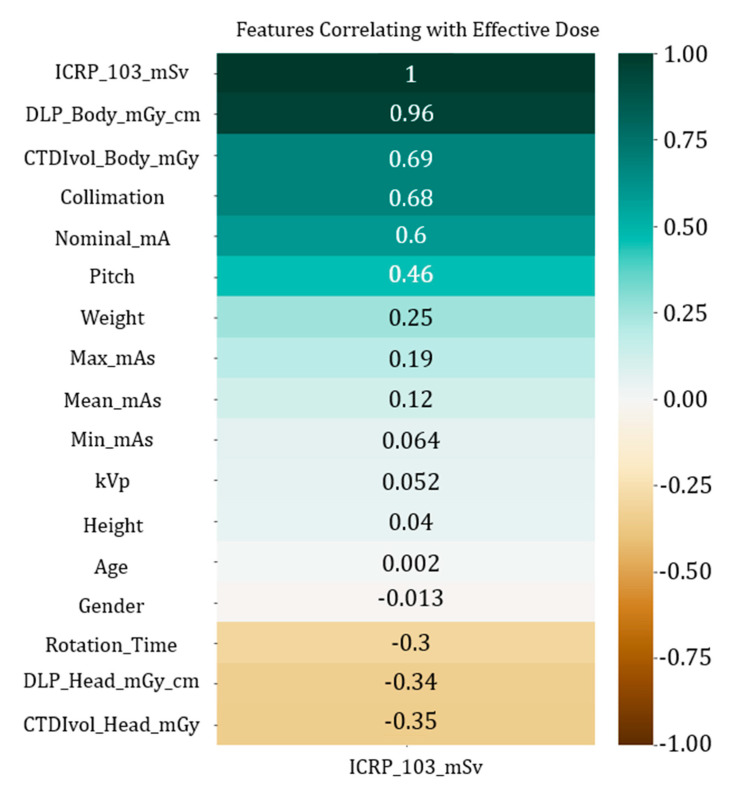
Correlation between features and target variable.

**Figure 4 tomography-11-00002-f004:**
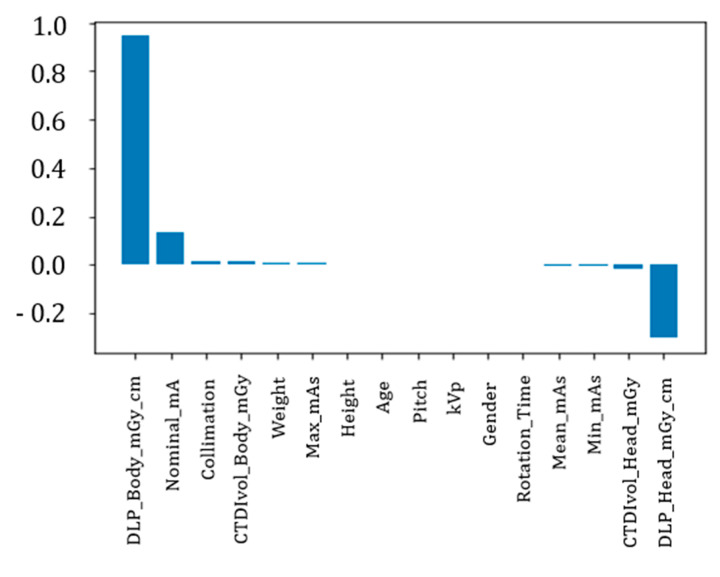
PCA importance (y axis) of the variable.

**Figure 5 tomography-11-00002-f005:**
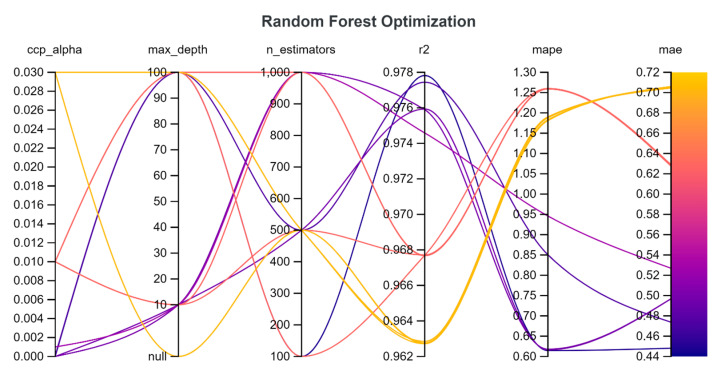
Example of optimization for random forest using Weights & Biases. Each configuration is a line that passes the point that represents the values of ccp_alpha, max_depth, and n_estimators chosen by the random search algorithm. R^2^, MAPE, and MAE are the results.

**Figure 6 tomography-11-00002-f006:**
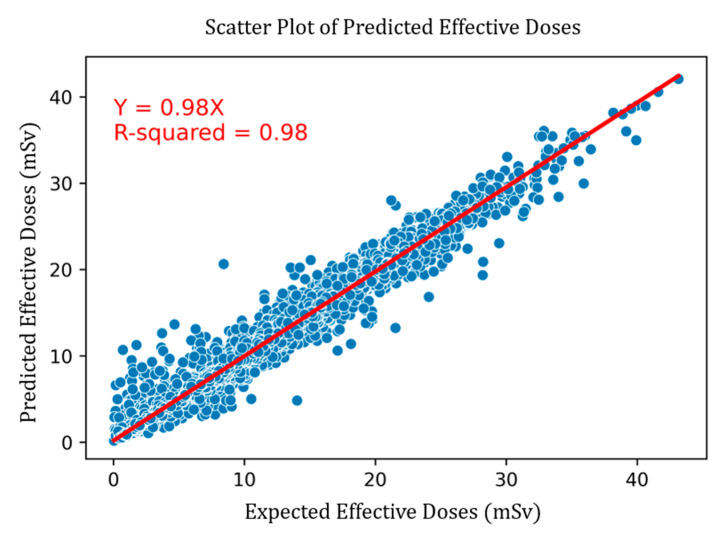
Predicted vs. expected doses for random forest regressor.

**Figure 7 tomography-11-00002-f007:**
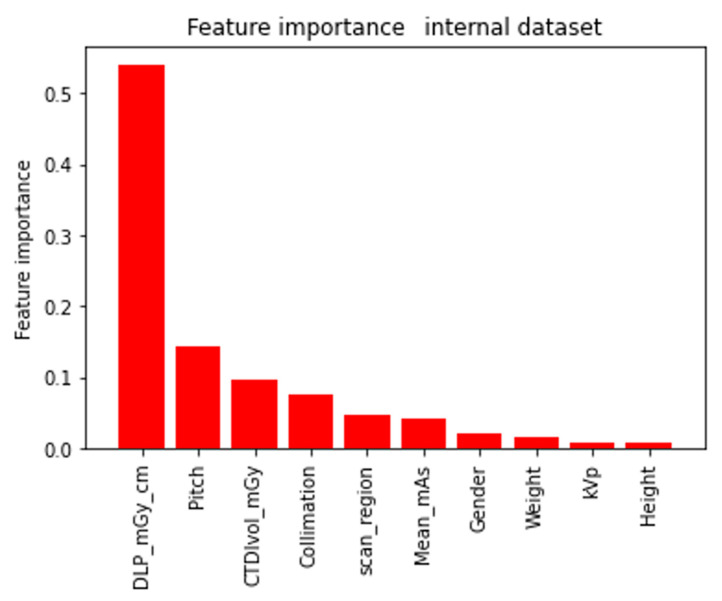
Feature importance for random forest regressor.

**Figure 8 tomography-11-00002-f008:**
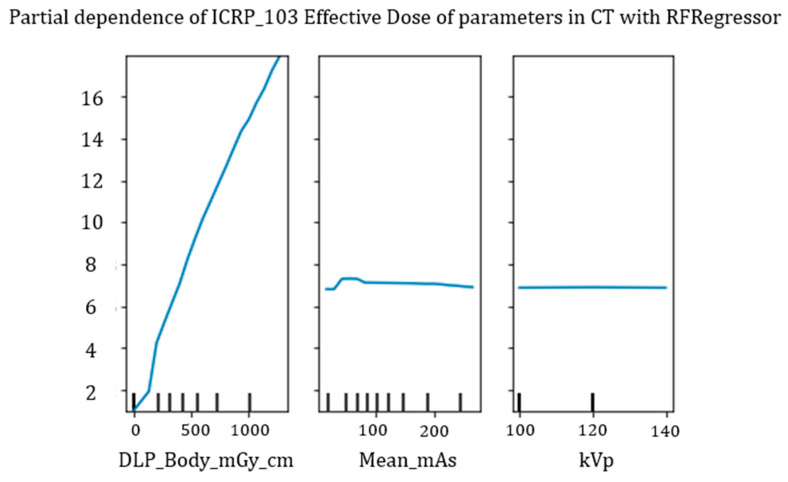
Dependence analysis for DLP, mean_mAs, and kVp.

**Table 1 tomography-11-00002-t001:** Possible configuration of the neural network that we used in our work in the optimization phase.

Hyperparameter	Choice 1	Choice 2	Choice 3	Choice 4
hidden_layers	[128]	[256, 128, 64]	[512, 256, 128, 64]	-
do	0	0.2	0.4	0.5
activation	relu	-	-	-
solver	adam	sgd	-	-
epochs	50	-	-	-
batch_size	256	512	1024	-
Init_lr	1 × 10^−2^	1 × 10^−3^	1 × 10^−4^	-

**Table 2 tomography-11-00002-t002:** Possible configuration for random forest regression.

Hyperparameter	Choice 1	Choice 2	Choice 3	Choice 4
n_estimators	100	500	1000	1 × 10^4^
max_depth	None	10	100	-
ccp_alpha	0	1 × 10^−4^	1 × 10^−3^	1 × 10^−2^

**Table 3 tomography-11-00002-t003:** The number of features used to feed the best-performing model, for the external validation.

6 Features	8 Features	10 Features
Gender	Gender	Gender
Weight	Weight	Weight
Height	Height	Height
CTDI_vol_ [mGy] DLP [mGycm]	CTDI_vol_ [mGy] DLP [mGycm]	CTDI_vol_ [mGy] DLP [mGycm]
Scan Region	Scan Region	Scan Region
	kV_p_	kV_p_
	Mean mAs	Mean mAs
		Collimation
		Pitch

**Table 4 tomography-11-00002-t004:** Summary of the results of different models on the internal test set.

Model	MAE [mSv]	MAPE [%]	R^2^
Neural Network	0.557	8.6	0.97
Support Vector Machine	0.658	23.3	0.96
Random Forest Regressor	0.416	7	0.98
k-factors and DLP	2.06	26	\
Multiple Linear Regression	0.98	44.4	0.93

**Table 5 tomography-11-00002-t005:** Results in dose class prediction from random forest regressor effective dose estimates.

DOSE Class	Precision	Recall	F1-Score	Support
I	1	0.99	0.99	3171
II	0.96	0.96	0.96	4904
III	0.93	0.93	0.93	4871
IV	0.96	0.96	0.96	4879
accuracy	0.96			17,735

**Table 6 tomography-11-00002-t006:** Results for the external validation as a function of the number of features.

	MAE (MAPE) 6 Features	MAE (MAPE) 8 Features	MAE (MAPE) 10 Features
External Validation—Turin	0.243 (7.9%)	0.224 (7.4%)	0.215 (7.1%)

Multiple linear regression for the external dataset resulted in a MAE of 0.89 mSv and a MAPE of 32.8% with an R^2^ of 0.74.

**Table 7 tomography-11-00002-t007:** Results in dose class prediction from the random forest regressor effective dose estimates for the external dataset.

DOSE Class	Precision	Recall	F1-Score	Support
I	0.96	0.80	0.87	205
II	0.96	0.98	0.97	2655
III	0.92	0.93	0.93	859
IV	0.93	0.85	0.89	81
Accuracy	0.95			3800

## Data Availability

Data are available upon reasonable request.

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
