# Peer review of "Effective Dose Estimation in Computed Tomography by Machine Learning"

_tomography, 2025, doi:10.3390/tomography11010002_

Round 1
Reviewer 1 Report
Comments and Suggestions for Authors
Dear authors, thank you for the possibility to review your manuscript.
I found it sound, well designed and of interest for the readers of this journal. Before publication, I have some suggestions to improve the manuscript that I would like to offer to the authors. Please find below my comments:
Introduction:
Minor: the introduction does not follow a standard structure: please re-write the final part of the introduction structuring as: A) Primary objectives, B) Secondary Objectives, and underlying hypothesis for both A and B.
Material and Methods:
Minor 2: I feel that the part mentioning the study design and pipeline should not be introduced before the data section, but it should have a dedicated paragraph. Also, you might consider drawing a figure that synthetize the study design including the machine learning approach used in the study.
Major 1: “We choose to feed our model with a subset of the parameters that are exportable from RadimetricsTM. In particular, we included in our model three different types of information” Why not using all the available variables?
Major 2: “We removed all samples that included missing values”.
Did you consider data imputation techniques. In machine learning it is common practice to exclude variables with more than 50% of missing values. For the remaining, data imputation techniques are suggested
Minor 3: “We also compute a correlation map (see Figure 1) between 119 features and target variable effective dose” .
Did you use Spearman or Pearson correlation. If the latter, can you then exclude non-linear dependencies?
Major 3: PCA analysis. How was the optimal number of components determined? How much variance was explained by each single component?
Results:
Internal validation
Minor 4: can the authors comment on how the different configuration settings (hyper parameters) would impact the performance of the algorithms? Are these results expected (as seen for example in Figure 4?
External validation
Major 4: I believe that the results presented for the DOSE class estimation should also be reported on the external dataset
Discussion
Minor1: the interpretability results should be moved to the section “Results”
Major2: the authors might consider the following studies and compare the results with the below references: [PMID: 31766708, https://doi.org/10.3390/app14031071]
Major 3: in the introduction the authors mention that the presented approach would allow a “real-time” estimate of the effective dose delivered to the patient. Given the authors’ expertise, I think they are in good position for suggesting how the presented algorithms would be implemented and the suggested infrastructure.
Author Response
Dear authors, thank you for the possibility to review your manuscript.
I found it sound, well designed and of interest for the readers of this journal. Before publication, I have some suggestions to improve the manuscript that I would like to offer to the authors. Please find below my comments:
Introduction:
Minor: the introduction does not follow a standard structure: please re-write the final part of the introduction structuring as: A) Primary objectives, B) Secondary Objectives, and underlying hypothesis for both A and B.
We want to thank Reviewer #1 for the valuable comments and feedback. The final part of the introduction has been rewritten, emphasizing the two aims of our work (page 2, line 54 in the clean version of the manuscript)
Material and Methods:
Minor 2: I feel that the part mentioning the study design and pipeline should not be introduced before the data section, but it should have a dedicated paragraph. Also, you might consider drawing a figure that synthetize the study design including the machine learning approach used in the study.
We added a figure to better explain the pipeline and the workflow used in our study. Figure has been added as Figure 1, the other figures have been renamed accordingly (page 2 line 69 in the clean version of the manuscript).
Major 1: “We choose to feed our model with a subset of the parameters that are exportable from RadimetricsTM. In particular, we included in our model three different types of information” Why not using all the available variables?
The reason we did not use all the available variables is related to the generalizability and exportability of our best model: the information related to patients, scanning parameters and dose indices are available and can be easily extracted from acquisitions performed with every device. The use of variables dose-tracking software specific (like for example the Size Specific Dose Estimates, a derived quantity) would have limited the applicability of our findings.
Major 2: “We removed all samples that included missing values”.
Did you consider data imputation techniques. In machine learning it is common practice to exclude variables with more than 50% of missing values. For the remaining, data imputation techniques are suggested
We did not consider data imputation technique, given the high number of samples in the cleaned dataset (over 69000 acquisitions).
Minor 3: “We also compute a correlation map (see Figure 1) between 119 features and target variable effective dose”.
Did you use Spearman or Pearson correlation. If the latter, can you then exclude non-linear dependencies?
We used Pearson correlation in order to evaluate linear (first order) dependencies, while non-linear dependencies should be evaluated and investigated with machine learning and deep learning approaches, as stated in the paper.
Major 3: PCA analysis. How was the optimal number of components determined? How much variance was explained by each single component?
Principal Component analysis was just an exploratory analysis, and we did not determine the optimal number of components, since all the models were fed with all the 16 features. In our PCA analysis, we found that CTDI_body and CTDI_Head accounted for 73% and 23% of the variance, respectively.
Results:
Internal validation
Minor 4: can the authors comment on how the different configuration settings (hyper parameters) would impact the performance of the algorithms? Are these results expected (as seen for example in Figure 4?
We added a brief part explaining that the results of the optimization phase are expected, given the hyperparameters and configuration chosen from machine learning algorithms and neural networks (page 12 line 337 in the clean version of the manuscript).
External validation
Major 4: I believe that the results presented for the DOSE class estimation should also be reported on the external dataset
We agree, and we added the results for dose class estimation in the “Results” section. We presented the values obtained with 10 features for the sake of simplicity: results for 6 or 8 features do not differ significantly (page 10 line 278 in the clean version of the manuscript).
Discussion
Minor1: the interpretability results should be moved to the section “Results”
Thank you for your feedback: the “Interpretability” paragraph has been moved at the end of the “Results” section (page 10 line 282 in the clean version of the manuscript).
Major2: the authors might consider the following studies and compare the results with the below references: [PMID: 31766708, https://doi.org/10.3390/app14031071]
We want to thank Reviewer #1 for the suggestion. However, we do not believe that results presented in those studies could be compared with our findings. One work deal with expected CTDI values according to patient BMI and scan region, while the other shows regression and correlation of dosimetric quantities (CTDI, DLP, effective dose and SSDE) starting from BMI Height and Weight. In particular, quantities that are used as dependent variables (CTDI and DLP), in our work are used as predictors of effective dose, thus the results are not comparable.
Major 3: in the introduction the authors mention that the presented approach would allow a “real-time” estimate of the effective dose delivered to the patient. Given the authors’ expertise, I think they are in good position for suggesting how the presented algorithms would be implemented and the suggested infrastructure.
We added a brief paragraph at the end of the “Discussion” section, to further emphasize this particular aspect (page 12 line 356 in the clean version of the manuscript).

Reviewer 2 Report
Comments and Suggestions for Authors
1. General comments
This is a study that compared multiple methods using machine learning and showed that the estimation of effective dose using machine learning is not inferior to conventional methods. The authors propose incorporating their method into conventional methods because it has a short calculation time. Although the method that uses only DLP has a large estimation deviation, conventional methods can also obtain prediction values that meet the objective with a small amount of calculation, so it is necessary to appeal more about the advantages of the method that uses machine learning. On the other hand, this research may be significant from the perspective of providing educational materials for learning machine learning.
2. Ethical concerns
A) This study uses patient data. For this reason, it is necessary to clearly state the ethical considerations of this study.
3. Introduction
A) Line 35: “According to ICRP 147 [1], effective dose is used in medicine for comparing doses from different medical procedures, informing judgements on justification, and establishing constraints for carers and volunteers in medical research.”
Is the uncertainty of the estimate required for this purpose consistent with the accuracy being pursued in this study?
4. Material and Methods
A) Line 131“To complete our exploratory analysis, we also included a PCA analysis to check which features affect mostly the variability in the output.”
Although the context is clear, it would be better to explicitly state that PCA is Principal component analysis.
B) Figure 2. Correlation between features and target variable
DLP_head and CTDI_head are negatively correlated with ICRP_103_mSv, and the contribution of DLP_head is second (Figure 3). Is this observation reasonable?
C) Figure 3. Correlation between features and target variable.
Although it is obvious from the title, it would be better if there was an explanation of the vertical axis.
5. Results
A) Figure 4
It would be better to add a figure that shows that optimization is being carried out using a random forest.
B) Figure 5. Predicted vs expected doses for random forest regressor.
It would be better to indicate that the units for the vertical and horizontal axes are mSv.
C) Line 242 “We fit all the selected models with the whole training dataset and then evaluated their performances using mean absolute error, mean absolute percentage error and r2.”
“R2” will be replaced with “coefficient determination”.
6. Discussion
A) Line 321: “For this purpose, we calculated the importance of the features,…”
It would be better to make it clear that this is an explanation of Figure 6.
B) Figure 6. Feature Importance for Random Forest Regressor.
It would be better if there were explanations of the vertical axis and IEO.
C) Line 330: “These plots are a measure of the trend of the prediction as the only variable indicated varies, perturbing the input data and plotting how the estimate varies according to the parameter studied.”
It would be better to make it clear that this is an explanation of Figure 7.
7. Acknowledgments
Since this study is made possible by the contributions of patients, it is better to mention that.
Author Response
- General comments
This is a study that compared multiple methods using machine learning and showed that the estimation of effective dose using machine learning is not inferior to conventional methods. The authors propose incorporating their method into conventional methods because it has a short calculation time. Although the method that uses only DLP has a large estimation deviation, conventional methods can also obtain prediction values that meet the objective with a small amount of calculation, so it is necessary to appeal more about the advantages of the method that uses machine learning. On the other hand, this research may be significant from the perspective of providing educational materials for learning machine learning.
- Ethical concerns
- A)This study uses patient data. For this reason, it is necessary to clearly state the ethical considerations of this study.
Given the retrospective nature of data, that were anonymized before the analysis, there was no need for approval by the Ethics Committee. In addition, we point out that CT dose indicators and exposure parameter data are mainly collected by dose tracking software for national legislation requirements, such as verification of diagnostic reference levels and also effective dose estimation for exposure indication in the examination report.
- Introduction
- A)Line 35: “According to ICRP 147 [1], effective dose is used in medicine for comparing doses from different medical procedures, informing judgements on justification, and establishing constraints for carers and volunteers in medical research.”
Is the uncertainty of the estimate required for this purpose consistent with the accuracy being pursued in this study?
Gold standard for the effective dose calculation is through Monte Carlo simulations on the actual CT images for a specific patient. Such approach, however, is impracticable in daily routine, and different approaches have been implemented.
Dose tracking softwares may use simplified MonteCarlo look up table to calculate organ and effective dose in population of phantoms, while k-factors applied to DLP derive from Monte Carlo simulations on different phantoms and specific body regions. This leads to a certain degree of variability in effective dose estimation: however, the methodology we used resulted in absolute differences from the expected values that are completely consistent with the purpose of the use of effective dose in medicine.
- Material and Methods
- A)Line 131“To complete our exploratory analysis, we also included a PCA analysis to check which features affect mostly the variability in the output.”
Although the context is clear, it would be better to explicitly state that PCA is Principal component analysis.
We explicitly state the meaning of PCA (page 4 line 133 in the clean version of the manuscript).
- B)Figure 2. Correlation between features and target variable
DLP_head and CTDI_head are negatively correlated with ICRP_103_mSv, and the contribution of DLP_head is second (Figure 3). Is this observation reasonable?
We believe this assumption is reasonable, since CTDI_Head and DLP_Head values don’t vary too much among different patients, thus providing little to no information regarding effective dose (ICRP_103_mSv).
- C)Figure 3. Correlation between features and target variable.
Although it is obvious from the title, it would be better if there was an explanation of the vertical axis.
Caption of figure 3 has been modified, we are sorry for the typo, since figure 3 represents the results of the PCA analysis (page 5 line 142 in the clean version of the manuscript).
- Results
- A)Figure 4
It would be better to add a figure that shows that optimization is being carried out using a random forest.
Figure 4 (now figure 5 in the clean version of the manuscript) represents the performance of Random Forest Regressor as a function of hyperparameters.
- B)Figure 5. Predicted vs expected doses for random forest regressor.
It would be better to indicate that the units for the vertical and horizontal axes are mSv.
Figure 5 has been replaced and units has been added in the vertical and horizontal axis (page 9 line 255 in the clean version of the manuscript, please note that now it is figure 6).
- C)Line 242 “We fit all the selected models with the whole training dataset and then evaluated their performances using mean absolute error, mean absolute percentage error and r2.”
“R2” will be replaced with “coefficient determination”.
The sentence has been rephrased according to the comment (page 8 line 246 in the clean version of the manuscript).
- Discussion
- A)Line 321: “For this purpose, we calculated the importance of the features,…”
It would be better to make it clear that this is an explanation of Figure 6.
Explicit reference to figure 6 has been added in the text (page 10 line 292 in the clean version of the manuscript).
- B)Figure 6. Feature Importance for Random Forest Regressor.
It would be better if there were explanations of the vertical axis and IEO.
Figure 6 has been replaced. The new figure contains the y-axis label, and “IEO” has been replaced with “internal dataset”, for the sake of clarity (page 11 line 298 in the clean version of the manuscript, please note that the figure is now figure 7).
- C)Line 330: “These plots are a measure of the trend of the prediction as the only variable indicated varies, perturbing the input data and plotting how the estimate varies according to the parameter studied.”
It would be better to make it clear that this is an explanation of Figure 7.
Reference to figure 7 has been made explicit in the revised version of the manuscript (page 11 line 302 in the clean version of the manuscript, please note that the figure is now figure 8).
- Acknowledgments
Since this study is made possible by the contributions of patients, it is better to mention that.
We want to thank Reviewer #2 for the suggestion, however we would point out that CT dose indicators and exposure parameter data are mainly collected by dose tracking software for national legislation requirements, such as verification of diagnostic reference levels and also effective dose estimation for exposure indication in the examination report.
